# High Rate of Mutations of Adhesion Molecules and Extracellular Matrix Glycoproteins in Patients with Adult-Onset Focal and Segmental Glomerulosclerosis

**DOI:** 10.3390/biomedicines11061764

**Published:** 2023-06-20

**Authors:** Sara Marcos González, Emilio Rodrigo Calabia, Ignacio Varela, Michal Červienka, Javier Freire Salinas, José Javier Gómez Román

**Affiliations:** 1Pathology Department, Marqués de Valdecilla University Hospital, Institute of Research Valdecilla (IDIVAL), 39008 Santander, Spain; 2Nephrology Department, Marqués de Valdecilla University Hospital, 39008, University of Cantabria, 39005 Santander, Spain; emilio.rodrigo@scsalud.es; 3Institute of Biomedicine and Biotechnology of Cantabria (IBBTEC), 39011, University of Cantabria-CSIC, 39005 Santander, Spain; ignacio.varela@unican.es; 4Nephrology Department, Rio Carrion General Hospital, 34005 Palencia, Spain; mcervienka@saludcastillayleon.es; 5Anatomic Pathology, Marqués de Valdecilla University Hospital, Institute of Research Valdecilla (IDIVAL), 39008 Santander, Spain; javifreiresalinas@gmail.com; 6Pathology Department, Marqués de Valdecilla University Hospital, Institute of Research Valdecilla (IDIVAL), School of Medicine, University of Cantabria, 39008 Santander, Spain; josejavier.gomez@scsalud.es

**Keywords:** adhesion molecules, glomerulonephritis, focal and segmental glomerulosclerosis, next-generation sequencing

## Abstract

(1) Background: Focal and segmental glomerulosclerosis (FSGS) is a pattern of injury that results from podocyte loss in the setting of a wide variety of injurious mechanisms. These include both acquired and genetic as well as primary and secondary causes, or a combination thereof, without optimal therapy, and a high rate of patients develop end-stage renal disease (ESRD). Genetic studies have helped improve the global understanding of FSGS syndrome; thus, we hypothesize that patients with primary FSGS may have underlying alterations in adhesion molecules or extracellular matrix glycoproteins related to previously unreported mutations that may be studied through next-generation sequencing (NGS). (2) Methods: We developed an NGS panel with 29 genes related to adhesion and extracellular matrix glycoproteins. DNA was extracted from twenty-three FSGS patients diagnosed by renal biopsy; (3) Results: The average number of accumulated variants in FSGS patients was high. We describe the missense variant *ITGB3*c.1199G>A, which is considered pathogenic; in addition, we discovered the nonsense variant *CDH1*c.499G>T, which lacks a Reference SNP (rs) Report and is considered likely pathogenic. (4) Conclusions: To the best of our knowledge, this is the first account of a high rate of change in extracellular matrix glycoproteins and adhesion molecules in individuals with adult-onset FSGS. The combined effect of all these variations may result in a genotype that is vulnerable to the pathogenesis of glomerulopathy.

## 1. Introduction

Focal and segmental glomerulosclerosis (FSGS) is one of the most common causes of adult-onset nephrotic syndrome alongside membranous nephropathy, and it is the primary glomerular disease that leads to end-stage renal disease (ESRD) in the United States [1,2]. Despite the numerous research studies conducted in recent years, an ideal therapy is still difficult to find, and a high percentage of patients eventually progress to ESRD because of years of unsatisfactory care and severe accompanying morbidity. In fact, 10 years following the diagnosis of primary FSGS, 45% of individuals develop ESRD [3].

Cell adhesion is essential in cell communication and regulation, being essential in the development and maintenance of tissues. Mechanical interactions between cells and between cells and their extracellular matrix (ECM) can influence and control cell behavior and function [4]. Cell adhesion status, involving both cell–cell and cell–matrix interactions, is a fundamental determinant of a wide variety of cell biological responses [5].

Maintenance of proper cell–cell adhesion is crucial for a variety of cell types. Cell–cell adhesion plays a vital role in homeostatic control by creating the physical tethering that permits the assembly of complex tissues. Furthermore, within individual cells, the activation of various signal transduction pathways is initiated by cell–cell adhesion receptors [5]. Within tissues, cells must adhere properly, not only to one another but also to components of the ECM that surrounds them [5]. Indeed, appropriate interactions between cells and the surrounding ECM are crucial for normal cellular survival. It is therefore logical that deregulation of cell–cell or cell–matrix interactions can contribute to the pathogenesis of human diseases [5,6].

Genetic research has contributed to the advancement of knowledge about FSGS syndrome, particularly in the case of congenital FSGS. More than 50 genes have been identified as being linked to hereditary FSGS, expanding the disease’s spectrum from children to some young adults [7]. Furthermore, various genetic studies have reported that certain genes, including *APOL1*, *WNK4*, *KANK1*, and *ARHGEF17*, are more prevalent in patients with primary FSGS [8,9]. The latter three genes alter the structure and function of podocytes, the key cells in the pathogenesis of FSGS, according to Yu et al.’s mouse model [9]. However, the exact mechanism by which *APOL1* variations contribute to FSGS is not well understood [9]. The initial stage leading to the segmental lesion involves the loss of podocytes and their detachment from the basement membrane [10].

The cytoskeleton maintains the structure and function of podocytes, but other molecules such as extracellular matrix glycoproteins and adhesion molecules collaborate on this task [11,12]. For example, among other adhesion molecules, some integrin mutations have been identified as being related to nephrotic syndrome [13]. In fact, treatment with the CTLA4-modulator abatacept restores β1 integrin activation of B7-1-expressing podocytes and reduces proteinuria in patients with B7-1-positive FSGS [14]. Unfortunately, abatacept therapy did not yield favorable results for several other patients who had male gender, FSGS, and post-transplant status, were children aged below 18 years, and who had previously failed treatment. These factors were associated with a lower remission rate, although the correlations did not attain statistical significance. This underscores the diverse mechanisms that contribute to the development of primary FSGS [15].

One advantage of the new recently available genetic research tools is that they allow the rapid analysis of multiple genes at the same time, enabling the discovery of previously unknown disease mechanisms. For example, next-generation sequencing (NGS) has been utilized to describe previously unidentified mutations in genetically complicated kidney illnesses such as steroid-resistant nephrotic syndrome, congenital anomalies of the kidney and urinary tract, and nephronophthisis [16,17]. Our hypothesis is that patients with primary FSGS may possess underlying changes in adhesion molecules or extracellular matrix glycoproteins that are associated with previously unreported mutations and that these can be investigated using NGS.

## 2. Materials and Methods

### 2.1. Patients and Controls

All the patients involved in this research were admitted and then followed up at the Nephrology Department of the Marqués de Valdecilla University Hospital. The study was conducted in accordance with the Declaration of Helsinki and approved by the Institutional Ethics Committee of Cantabria (protocol internal code 2019.206; date of approval: 4 October 2019).

We analyzed the histological results after the pathologist in charge of the nephropathology division had routinely diagnosed all renal biopsy slides.

We included twenty-three patients with FSGS that had been demonstrated by renal biopsy during the period 2004–2020 and followed them clinically for a minimum of 2 years. The main patient characteristics are shown in Table 1. Briefly, the mean age was 53.96 ± 18.81 years, 26% were female, and 74% suffered high blood pressures. The mean estimated glomerular filtration rate was 60.30 ± 29.95 mL/min, mean proteinuria was 10.13 ± 5.96 g/day, and 43% showed nephrotic syndrome.

A total of 16 samples of normal renal tissue received at the Pathology Department of the Marqués de Valdecilla University Hospital in Santander were selected as the control group; these corresponded to renal wedges (from organ donors) and radical nephrectomies (mainly due to renal cancer) of patients with preserved renal function. Subsequently, after a thorough review of the criteria, the control group comprised 15 samples, as one sample was not selected due to a diagnosis of IgA GN at another hospital. 

In addition, CNVs were compared with publicly available control databases, such as the Genome Aggregation Database (gnomAD), the Database of Genomic Variants (DGV), and databases for pathogenic CNVs, such as ClinVar.

### 2.2. Identification and Annotation of Genes of Cellular Adhesion Molecules

Using information gathered from a variety of sources, including databases such as the National Center for Biotechnology Information (NCBI) and journal publications, 29 genes were chosen as potential human cell-adhesion molecule gene candidates [4]: *ACTB, ARHGDIA CDC42, CDH1, COL1A2, ELN, FN1, FSCN1, ICAM1, ITGA5, ITGB1, ITGB3, LAMC1, LPAR1, LPAR2, LPAR3, LPAR4, LPAR5, LPAR6, PTK2, RHOA, RHOC, ROCK2, SELE, TNC, TPM1, TRAM1, VCAM1,* and *WASF3*.

Based on Online Mendelian Inheritance in Man (OMIM; https://www.omim.org, accessed on 4 March 2022), we distinguished between nephropathic and phenocopy genes. As a result, “nephropathic genes” were defined as genes that were recognized in OMIM as causing “nephropathy”. In contrast, “phenocopy genes” were those genes in OMIM found to produce a syndromic or unrelated illness. (Appendix A). Where a case was not yet documented in OMIM, recent research pointing to the gene’s causal pathogenic role in the nephrotic syndrome was considered to stratify cases as nephropathic vs. phenocopy [18,19].

### 2.3. Sample Preparation and Mutational Analysis

The renal biopsy was performed by percutaneous strategy under ultrasound guidance. Renal tissues were processed, formalin fixed, and paraffin embedded according to standard procedures for histological diagnosis.

Genomic DNA was extracted from formalin-fixed paraffin-embedded (FFPE) material. From the FFPE material, 5 sections of 3 µm were taken and collected in a 1.5 mL Eppendorf tube. First, tissue deparaffinization was carried out by adding 160 µL of Deparaffinization Solution (QIAGEN, Hilden, Germany) and incubating at 56 °C for 5 min at 700 rpm. Next, DNA extraction was carried out using Cobas^®^ DNA Sample Preparation Kit (Roche Diagnostics, Basel, Switzerland) according to the manufacturer’s protocol (Appendix A).

Once DNA was extracted from the samples, nucleic acid quantification was performed using the Qubit 1X dsDNA HS Assay Kit (ThermoFisher Scientific, Waltham, MA, USA); according to the manufacturer, samples at a concentration of at least 2 µg/uL were considered valid for sequencing. Finally, all the samples were diluted up to 1 ng/µL to begin the preparation of libraries.

#### 2.3.1. Design of the Targeted Sequencing Panel

We designed a custom panel for the targeted sequencing of DNA extracted from FFPE material using AmpliSeq technology (ThermoFisher Scientific), which is the standard for FFPE sequencing.

Due to the nature and quality of the FFPE material, PCR amplification was performed instead of hybrid capture. We designed a panel with complete exon coverage +/− 15 bases for possible splicing and leaving the UTR control regions unsequenced for the following genes: *ACTB*, *ARHGDIA*, *CDC42*, *CDH1*, *COL1A2*, *ELN*, *FN1*, *FSCN1*, *ICAM1*, *ITGA5*, *ITGB1*, *ITGB3*, *LAMC1*, *LPAR1*, *LPAR2*, *LPAR3*, *LPAR4*, *LPAR5*, *LPAR6*, *PTK2*, *RHOA*, *RHOC*, *ROCK2*, *SELE*, *TNC*, *TPM1*, *TRAM1*, *VCAM1*, and *WASF3.* All this was carried out using an automated platform, Ion AmpliSeq On-Demand Panels for Targeted Sequencing (ThermoFisher Scientific), which formulates the necessary primers.

Primer design was performed for the FFPE tissue to generate 175 bp length amplicons, covering 92.65 kb over 29 genes with a total 910 amplicons.

#### 2.3.2. Library Preparation

Library preparation was performed automatically using the Ion AmpliSeq kit for Ion Chef DL8 (ThermoFisher Scientific), with a 31-amplification-cycle program of 4 min. Once amplified, libraries were quantified and diluted to equimolar concentrations.

#### 2.3.3. Sequencing

Sequencing was performed using Ion S5 (ThermoFisher Scientific) with a 520 chip, which generated a minimum of 5 million useful reads to determine germinal coverage to at least ×100.

#### 2.3.4. Sequencing Data Collection

Only cases with a coverage greater than or equal to 160 reads were selected for analysis. In those patients where the library had a depth of less than 160 reads, the entire sequencing process was repeated.

#### 2.3.5. Bioinformatic Analysis

The human genome (hg19) was aligned using BWA (http://bio-bwa.sourceforge.net/, accessed on 18 June 2020), which was used to align fastq files. Later, the alignments were sorted and filtered using samtools and picardtools (https://broadinstitute.github.io/picard**/** and http://www.htslib.org/, respectively) (accessed on 18 June 2020). UnifiedGenotyper was used to detect mutations (https://gatk.broadinstitute.org/hc/en-us/articles/360036365812-HaplotypeCaller, accessed on 18 June 2020). Using vcftools (http://vcftools.sourceforge.net/, accessed on 18 June 2020), we then filtered the mutations in the region of interest. PED and MAP files were created using customized PERL scripts to perform association studies with the Plink program (https://www.cog-genomics.org/plink/1.9/, accessed on 18 June 2020); default values were utilized without considering gender differences, and multitest correction was not performed. Finally, annotation of the functional consequence of each mutation was carried out using custom PERL scripts using the Ensembl database API.

All detected alterations were classified according to the ACMG guideline [20,21] using the Franklin ACMG Classification (https://franklin.genoox.com/clinical-db/home, accessed on 1 June 2023).

## 3. Results

### 3.1. Clinical and Pathological Characteristics

A total of 23 patients were included in the study (clinical and histopathological characteristics are summarized in Table 1). Of these patients, 74% were male, and 26% were female. The median age was 54 years (range 18–82 years); mean age: 54 years), and all 23 patients were adults ≥18 years old.

In almost all cases (74%; *n* = 17), classical or NOS (not otherwise specified) FSGS was the predominant morphological variant observed on biopsy. A perihilar variant was identified in four patients (17%; *n* = 4), while cellular and tip variants were observed in one patient each.

Patients with diabetic nephropathy or hypertensive nephropathy were excluded from the study. Patients were classified into chronic kidney disease (CKD) stages according to the established KDIGO classification [22]: 30% stage 1, with normal or high GFR (≥90 mL/min); 18% stage 2, mild CKD (60–89 mL/min); 9% stage 3a (45–59 mL/min); 30% stage 3b (30–44 mL/min); and 13% stage 4, severe CKD (15–29 mL/min); no patients were detected in stage 5.

### 3.2. Mutational Analysis

Of the 29 genes studied, variants were found in all of them, except in the *RHOA* and *LPAR5* genes, although many of them were silent mutations (Appendix A). UTR regions as well as non-coding regions, which do not affect splicing, were not considered. All the raw sequencing data can be accessed from the European Nucleotide Archive (ENA) using this link: https://www.ebi.ac.uk/ena/data/search?query=PRJEB53199 (accessed on 31 May 2022).

Initially, we examined all non-silent mutations, specifically noting the presence of alterations in the *FN1* and *TNC* genes in all FSGS patients. The frequencies of variations in the different genes analyzed are shown in Figure 1.

Variations were observed in both the control group and in all patients, although the average number of cumulative variants (Single Nucleotide Polymorphisms, SNPs) was higher in FSGS patients (16.5 SNPs) compared with the control group (mean of 11.8 SNPs).

We then shifted our focus to non-benign variants; specifically, those classified as pathogenic, probably pathogenic, and uncertain significant variants (Table 2). Two patients were found to have the pathogenic missense variant *ITGB3*c.1199G>A, and five FSGS patients were discovered to have the likely pathogenic nonsense variant *CDH1*c.499G>T, which has no Reference SNP (rs) Report. The missense variants *COL1A2*c.1015A>C, *ITGB1*c.1807A>T, *ITGB3*c.2351C>T, *LAMC1*c.148T>C, *LPAR3*c.524C>G, *LPAR4*c.259C>G, *LPAR4*c.260T>C, *LPAR6*c.227A>T, *LPAR6*c.998T>C, and *WASF3*c.934G>C were classified as uncertain significant variants. None of the gene variants listed above were detected in the control group, but the difference was not statistically significant.

Five patients had the *CDH1*c.499G>T mutation (patients 1, 2, 5, 7, and 9): two women and three men, with a mean age of 52.6 years. Two of them also had the *ITGB3*c.1199G>A mutation (patients 2 and 7). The first patient was diagnosed with autoimmune hemolytic anemia and is currently being studied for interstitial lung disease (ILD) and receiving treatment with Rituximab. The other patient developed an advanced chronic disease with subsequent transplantation and recurrence of FSGS in the control biopsy. The rest of the patients with the *CDH1*c.499G>T mutation are currently under control without immunosuppressive treatment and have mild CKD.

The remaining variants were deemed benign, with many having similar frequencies in the overall European population (Appendix A). Additionally, we identified new variants that have not yet been annotated and therefore have no Reference SNP (rs) Report (Table 3).

On the other hand, when we searched for a correlation between non-benign mutations and disease severity, as measured by the level of proteinuria, we observed a tendency toward a greater accumulation of mutations in the subnephrotic range of proteinuria (Figure 2).

Furthermore, to directly correlate these mutations with glomerulopathies, we considered the stages of CKD. According to the statistical analysis, the number of mutations is not dependent on the stage of chronic kidney disease (*p* = 0.1164). However, on average, stage 3a has 2.42 more mutations compared with stage 1. We created a graph showing the distribution of mutations by genes and CKD stages (Figure 3).

## 4. Discussion

In our cohort of patients with adult-onset FSGS, we observed for the first time a high rate of mutations in adhesion molecules and extracellular matrix glycoproteins, with the high prevalence of non-silent mutations the key result of our investigation, although the limited sample size does not allow for statistical significance. Both in the general population (European population, dbSNP, 1000 Genomes) and in our control group, this rate of alteration is lower than in the FSGS patient group (Appendix A). Similar to this study, other genetic studies have previously reported mutations in adhesion molecules and extracellular matrix glycoproteins in patients with FSGS [23,24,25].

Most genetic research has established a direct correlation between mutations in podocyte and collagen *COL4A* (*A3/A4/A5*) genes and the development of familial FSGS. This correlation persists even in cases of adult-onset FSGS, where a monogenic cause of FSGS could be identified in up to 29% of cases [7,8,9,26,27]. A similar rate of mutation was reported in adult-onset steroid-resistant nephrotic syndrome, with a higher age of onset of FSGS related to a lower rate of single-gene identification [28].

Mutations in the *INF2* gene have been reported in familial FSGS of both Caucasian and Asian ancestry [23,24], while Marx et al. identified a novel nonsense variant in the *PODXL* gene in a three-generation family with an atypical glomerular nephropathy resembling FSGS [25]. Although these studies found mutations in specific molecule groups, they did not investigate the entire set of genes that constitute these groups.

To perform a comprehensive analysis of these genes, we utilized NGS technology, which allowed us to rapidly study multiple genes in a single experiment at high resolution [16]. This approach facilitates the identification of new molecules or pathways involved in or contributing to the development of FSGS. However, interpreting new variants without a known clinical significance can be challenging, and additional functional tests are necessary to link these variants to FSGS [7,16,23].

It is noteworthy that in our study, all the FSGS patients showed alterations in tenascin and fibronectin 1, which have previously been linked to podocyte injury [29,30,31]. Additionally, a large percentage of our patients exhibited mutations in integrins (78%) and laminins (87%), which have been described in other studies [32,33] that we will discuss below.

While podocyte degradation is a critical step in the pathophysiology of FSGS, the environment may also contribute to the development of segmental lesions. Research has shown that alterations in adjacent epithelial parietal and mesangial cells also play a role in FSGS [34]. Extracellular matrix glycoproteins and adhesion molecules also contribute to podocyte health [11,29,35,36], and animal and cell culture studies have linked alterations in these molecules to nephrotic syndrome and podocyte damage [30,31,32,37,38,39,40]. Moreover, mutations in adhesion proteins such as laminin β2, integrin α3, and integrin β4 in humans have been associated with steroid-resistant nephrotic syndrome due to a lack of appropriate adhesion of podocytes to the glomerular basement membrane [13,41,42,43]. Experimental and human FSGS exhibit altered expression of extracellular matrix proteins, including laminin-1, perlecan, collagen type IV-2, laminin-2, agrin, and collagen type IV-4, that are produced by parietal epithelial cells and podocytes [32]. Therefore, these variations in extracellular matrix proteins may possibly have an impact on the histopathologic type of FSGS [44].

In addition to other signals, adhesion molecules and extracellular matrix glycoproteins are components of a network that involves redundant interactions between molecules [29]. The heterogeneity of research and the difficulty in identifying a single common etiology of a disease that is frequently syndromic is explained by this complicated network [26,45]. 

It is important to note that, in our study, no mutation in a single gene was detected that could explain FSGS, but each patient had a variable number of non-silent mutations (9 to 22 non-silent mutations per patient) (Appendix A), which could partially explain the predisposition to the disease.

Multiple gene variants may interact to promote podocyte damage caused by different non-Mendelian forms of FSGS [9]. Therefore, each individual mutation affects only a few patients, and the addition of mutations that may promote these minor injuries contributes to the development of segmental lesions. In this sense, a greater number of mutations in adhesion molecules and extracellular matrix glycoproteins could favor the occurrence of FSGS by increasing susceptibility to the disease [9]. It is interesting that several authors have noted a synergistic effect of various FSGS mutations. Frese et al. demonstrated that carriers of type IV collagen (*COL4A5*) gene mutations with related polymorphisms in the slit diaphragm genes experience severe forms of FSGS [46], while Bullich et al. report a similar result with *COL4A3* mutations [17].

Notably, we cannot be sure of the exact role of the variants we found, and we classified most of them as benign or “of uncertain significance” [21], understanding that further studies will be needed to understand the pathogenic role of adhesion molecule and extracellular matrix glycoprotein mutations in the development of FSGS.

Several limitations should be mentioned. Although our study sheds light on the prevalence of mutations in adhesion molecules and extracellular matrix glycoproteins in adult-onset FSGS patients, it is limited by a small sample size. Furthermore, the lack of a well-defined classification system for adult-onset FSGS and the absence of validated methods for distinguishing non-familial FSGS [22] make it difficult to draw definitive conclusions. While we excluded patients with secondary forms of FSGS and suspected familial forms, it is still possible that some patients had these forms. Additionally, the absence of electron microscopy analysis prevented us from differentiating between primary and adaptive forms of FSGS. Further research is needed to fully understand the complex pathogenesis of this heterogeneous disease.

## 5. Conclusions

We were able to identify a high rate of mutation in adhesion molecules and extracellular matrix glycoproteins in all of the selected adult patients with FSGS.

We found mutations without a Reference SNP (rs) Report. Two patients were found to have the pathogenic missense variant *ITGB3*c.1199G>A, and five FSGS patients were discovered to have the likely pathogenic nonsense variant *CDH1*c.499G>T. None of the gene variants were detected in the control group; however, this difference was not statistically significant.

## Figures and Tables

**Figure 1 biomedicines-11-01764-f001:**
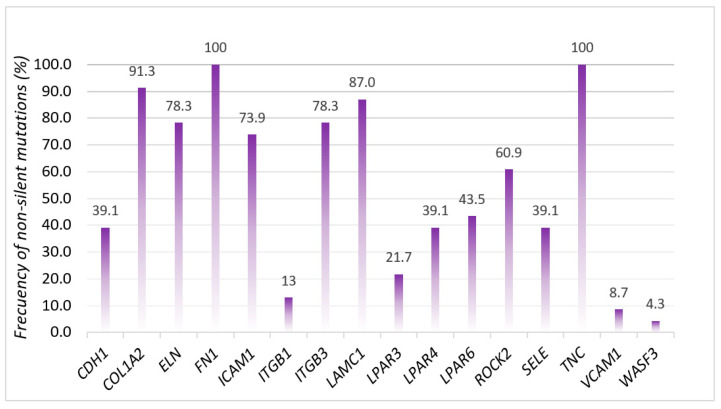
Frequencies of non-silent mutations (%).

**Figure 2 biomedicines-11-01764-f002:**
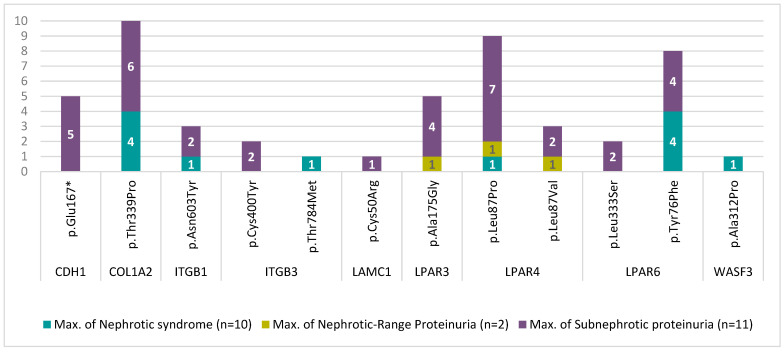
Distribution of non-benign mutations according to disease severity (proteinuria).

**Figure 3 biomedicines-11-01764-f003:**
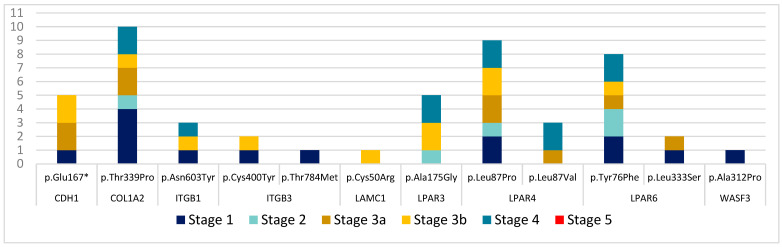
Frequency of mutations by genes and CKD stages.

**Table 1 biomedicines-11-01764-t001:** Clinical and pathological characteristics of patients with focal and segmental glomerulosclerosis included in the study.

Patients	Clinical Characteristics		Pathological Characteristics	Clinical Onset					
	Gender	Age	Morphological Variants	Proteinuria (g/Day)	Serum Albumin (g/dL)	HBP	Creatinine (mg/dL)	Creatinine Clearance (mL/min)	CKD Stage
1	M	52	Perihiliar	2.2	3.1	Yes	1.9	40	3b
2	F	68	NOS	3.0	3.7	Yes	0.7	91	1
3	M	82	NOS	2.1	3.4	Yes	0.9	80	2
4	M	26	Perihiliar	4.3	3.1	No	1.5	110	1
5	M	68	NOS	1.3	4.2	Yes	1.5	46	3a
6	M	52	NOS	2.6	3.2	Yes	1.8	42	3b
7	M	38	NOS	0.4	3.8	No	2.2	37	3b
8	M	70	NOS	6.5	3.8	Yes	1.9	35	3b
9	F	37	NOS	1.3	4.1	Yes	1.3	49	3a
10	M	79	NOS	1.2	3.6	Yes	2.4	28	4
11	M	65	NOS	6.5	3.6	Yes	3.8	17	4
12	M	56	NOS	3.8	3.4	Yes	1.0	60	2
13	M	62	NOS	6.6	3.1	Yes	2.8	25	4
14	M	43	NOS	6.3	3.0	Yes	1.8	60	2
15	M	56	NOS	5.5	3.3	No	0.9	100	1
16	F	74	Tip lesion	1.6	3.8	No	1.0	90	1
17	M	54	NOS	12.9	2.8	Yes	0.9	100	1
18	M	73	Cellular	1.6	2.8	Yes	3.3	30	3b
19	M	53	NOS	6.5	3.0	Yes	0.9	75	2
20	F	67	NOS	32.0	1.9	Yes	1.4	37	3b
21	F	29	Perihiliar	6.2	3.5	No	2.3	35	3b
22	F	18	NOS	1.9	4.0	No	1.2	100	1
23	M	19	Perihiliar	15.0	2.7	Yes	0.8	100	1

F, female; M, male; HBP, high blood pressure; CKD, chronic kidney disease.

**Table 2 biomedicines-11-01764-t002:** The genetic data of patients with FSGS according to sequencing, based on the Franklin ACMG guideline.

Gene	Variant	Frequency in Patients (*n* = 23)	Frequency in European Population (dbSNP, 1000 G)	Zygosity	ACMG
*CDH1*	c.499 G>T (p.Glu167 *)	(5) 0.2174		Het	LP
*COL1A2*	c.1015 A>C (p.Thr339Pro)	(9) 0.3913		Het	US
*ITGB1*	c.1807 A>T (p.Asn603Tyr)	(3) 0.1304		Het	US
*ITGB3*	c.1199 G>A (p.Cys400Tyr)	(2) 0.0869	0.0000	Het	P
	c.2351 C>T (p.Thr784Met)	(1) 0.0435	0.0010	Het	US
*LAMC1*	c.148 T>C (p.Cys50Arg)	(1) 0.0435		Het	US
*LPAR3*	c.524 C>G (p.Ala175Gly)	(5) 0.2174		Het	US
*LPAR4*	c.260 T>C (p.Leu87Pro)	(9) 0.3913		Het	US
	c.259 C>G (p.Leu87Val)	(3) 0.1304		Het	US
*LPAR6*	c.227 A>T ( p.Tyr76Phe)	(7) 0.3043		Het	US
	c.998 T>C (p.Leu333Ser)	(2) 0.0869		Het	US
*WASF3*	c.934 G>C (p.Ala312Pro)	(1) 0.0435		Het	US

ACMG, American College of Medical Genetics and Genomics; Het, heterozygous; LP, likely pathogenic; P, pathogenic; US, uncertain significance; *CDH1*, cadherin 1; *COL1A2*, collagen type I alpha 2 chain; *ITGB1*, integrin subunit beta 1; *ITGB3*, integrin subunit beta 3; *LAMC1*, laminin subunit gamma 1; *LPAR3*, lysophosphatidic acid receptor 3; *LPAR4*, lysophosphatidic acid receptor 4; *LPAR6*, lysophosphatidic acid receptor 6; *WASF3*, WASP family member 3; *, STOP codon.

**Table 3 biomedicines-11-01764-t003:** Mutations found without Reference SNP (rs) Report.

Gene	rs	Nucleotide Change	Amino Acid Change	Type of Alteration	ACMG
*CDH1*	-	c.500 A>G	p.Glu167Gly	Missense	LB
	-	c.499 G>T	p.Glu167 *	Nonsense	LP
*COL1A2*	-	c.1015 A>C	p.Thr339Pro	Missense	US
*ELN*	-	c.1435 G>A	p.Val479Met	Missense	LB
*ICAM1*	-	Exon 3 + 1 G>A	-	Essential splice	-
*ITGB1*	-	c.1807 A>T	p.Asn603Tyr	Missense	US
*LAMC1*	-	c.148 T>C	p.Cys50Arg	Missense	US
*LPAR3*	-	c.524 C>G	p.Ala175Gly	Missense	US
*LPAR4*	-	c.259 C>G	p.Leu87Val	Missense	US
	-	c.260 T>C	p.Leu87Pro		US
*LPAR6*	-	c.227 A>T	p.Tyr76Phe	Missense	US
	-	c.998 T>C	p.Leu333Ser		US
*TNC*	-	c.4241 G>C	p.Arg1414Thr	Missense	LB
	-	Exon 28—2 A>T	-	Essential splice	-
*WASF3*	-	c.934 G>C	p.Ala312Pro	Missense	US

ACMG, American College of Medical Genetics and Genomics; LB, likely benign; LP, likely pathogenic; P, pathogenic; US, uncertain significance; *CDH1*, cadherin 1; *COL1A2*, collagen type I alpha 2 chain; *ELN*, elastin; *ITGB1*, integrin subunit beta 1; *ITGB3*, integrin subunit beta 3; *LAMC1*, laminin subunit gamma 1; *LPAR3*, lysophosphatidic acid receptor 3; *LPAR4*, lysophosphatidic acid receptor 4; *LPAR6*, lysophosphatidic acid receptor 6; *TNC*, Tenascin C; *WASF3*, WASP family member 3; *, STOP codon.

## Data Availability

https://www.ebi.ac.uk/ena/data/search?query=PRJEB53199 (accessed on 31 May 2022).

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
