# Peer review of "High Rate of Mutations of Adhesion Molecules and Extracellular Matrix Glycoproteins in Patients with Adult-Onset Focal and Segmental Glomerulosclerosis"

_biomedicines, 2023, doi:10.3390/biomedicines11061764_

Round 1

Reviewer 1 Report

The study is interesting, and the way it is conducted, I mean the methodology, is correct. My main doubt stems from the authors showing numerous mutations, but what exactly results from it? I read the discussion carefully and didn't find an answer there.
It would be helpful to divide the discussion into sections, group the mutations, and address them.
Perhaps a broader introduction to the role of extracellular matrix glycoproteins and adhesion molecules would be helpful.

Are the authors sure that these mutations are related to FSGS? Maybe a result of ESRD. Is it possible to compare the results from patients with FSGS with patients with ESRD, e.g., due to other kidney diseases, then, it would be clear what results from FSGS. Is it known, for example, whether the patients had diabetic nephropathy or hypertensive nephropathy? I didn't find this in the exclusions for the study.
The conclusions do not convince me; they are very general.

The English language in the article is poor; many sentences are grammatically and stylistically weak, and there are repetitions as if the authors have not reread the article. For example: "Only pathogenic, likely pathogenic, and uncertain significant variants were shown here according to the ACMG guideline."

Author Response

An extensive language editing has been carried out (reviewed by a native speaker), but in case more is required, it has been agreed to send the manuscript to the journal's own editing service. Unnecessary repetitions have also been removed.

The introduction has been expanded with respect to the role of extracellular matrix glycoproteins and adhesion molecules to better contextualize the study.

The results are presented in a more organized manner, as well as the discussion section, which addresses the results obtained by sections.

To answer this question: "Are the authors sure that these mutations are related to GEFS?" we have considered the stages of chronic kidney disease and, according to statistical analysis, the number of mutations is not dependent on the stage of chronic kidney disease. We have added this to the results section.

We added in the results section the specification of excluding patients with diabetic nephropathy or hypertensive nephropathy.

Although the conclusions are general, I have tried to adapt them to our study and develop them further in the discussion section.

Reviewer 2 Report

The authors studied mutations in adhesion molecules and ECM glycoproteins in patients with FSGM using NGS.

Comments

1.      The language of the paper should be improved. The scientific word use should be applied. All the typos should be corrected.

2.      Lines 29-30; 48-50; 60-61;142; 198-200; 204-205; 207-209: These sentences are not clear. They should be clarified.

3.      Lines 39-41; 52-61: These pieces should be rephrased as they are not clear.

4.      Lines 67-69: The characteristics of patients who did not response to treatment should be described.

5.      Section 2.1: The number of patients used in the study is critically low for genetic studies. Therefore, the results and conclusions might be not meaningful.

6.      Lines 115-119: The protocol should be described in more detail.

7.      Lines 163-164; 168-170: The authors should present the criteria for their decision that the variant is pathogenic.

8.      Lines 219: The reference is required at the end of this sentence.

9.      Line 257: COLA3 does not exist. This should be corrected.

10.  Line 255: Correct writing is type IV collagen (COL4A5)

11.  References should be rearranged according to the Journal requirements.

The language of the paper should be improved. The scientific word use should be applied. 

Author Response

The introduction has been expanded regarding the topic of the role of extracellular matrix glycoproteins and adhesion molecules, in order to better contextualize the study, with corresponding references.

The materials and methods section has been extended and explained in greater detail. Additionally, the results are presented in a more organized manner.

1. The language of the paper should be improved. The scientific word use should be applied. All the typos should be corrected.

2. Lines 29-30; 48-50; 60-61;142; 198-200; 204-205; 207-209: These sentences are not clear. They should be clarified.

3. Lines 39-41; 52-61: These pieces should be rephrased as they are not clear.

(1,2,3) Regarding language, an extensive language editing has been performed (reviewed by a native speaker), and if further improvement is needed, it has been agreed to send the manuscript to the journal's editing service. The indicated sentences have been rewritten.

  1. Lines 67-69: The characteristics of patients who did not response to treatment should be described.

We added the suggestions of the study of non-response to treatment.

  1. Section 2.1: The number of patients used in the study is critically low for genetic studies. Therefore, the results and conclusions might be not meaningful.

The study limitation due to the low number of patients and controls is specified, noting that the results are not statistically significant, and there is a need to increase the number of participants (in the discussion and conclusion sections).

  1. Lines 115-119: The protocol should be described in more detail.

The materials and methods section has been extended and explained in greater detail, emphasizing the followed protocols.

  1. Lines 163-164; 168-170: The authors should present the criteria for their decision that the variant is pathogenic.

We added the criteria for the decision that the variant is pathogenic: Franklin ACMG Classification (https://franklin.genoox.com/clinical-db/home)

  1. Lines 219: The reference is required at the end of this sentence. The reference is not added to the end of this sentence. The sentence is modified because the data corresponds to our cohort, and it is added to results too.
  2. Line 257: COLA3 does not exist. This should be corrected.

Corrected the writing COL4A3.

  1. Line 255: Correct writing is type IV collagen (COL4A5)

Modification writing type IV collagen (COL4A5)

  1. References should be rearranged according to the Journal requirements.

All references are rearranged according to journal requirements.

Round 2

Reviewer 1 Report

The authors read my comments well and corrected the article accordingly. In my opinion it is now good and may be accepted and considered for publication.

Author Response

Thank you very much for the contributions received.

Reviewer 2 Report

The authors tried to improve their manuscript but more work is required.

Comments

1.      Line 67: Reference is required at the end of the sentence.

2.      Lines 76-78,86-88, 185-186, 203-204, 295-296: These sentences are not clear. They should be rephrased.

3.      Lines 98-99: Short general characteristic of the whole cohort should be presented.

4.      Line 125, 126, etc: Reference and a description “In brief” should be presented for each of the methods used in the study.

5.      Lines 129,132,138, 147, 156, 159: For each kit or reagent the authors should indicate the Company, City, and Country where it was purchased.

6.      Lines 188-190: The detailed analysis of patients’ clinical data should be presented. The authors should also describe the differences between different SKD stages.

7.      Lines 218-221: The authors should describe specific details of 2 patients with ITGB and 5 patients with CDH1 where the most important mutations were found.

8.      Line 266: It is not clear which COL4A gene and collagen are described. This should be corrected.

9.       Limitations cannot be considered as Conclusions. Limitations should be moved to the Discussion section.

10.  Conclusions are missing. Conclusions should describe the authors findings.

Many sentebces are not clear. The manuscript is required extensive editing.

Author Response

  1. Line 67: Reference is required at the end of the sentence.

Reference added at the end of the sentence.

  1. Lines 76-78,86-88, 185-186, 203-204, 295-296: These sentences are not clear. They should be rephrased.

The unclear sentences that were indicated are rewritten.

The unclear sentences that were indicated are rewritten.

76-81. Unfortunately, abatacept therapy did not yield favorable results for several other patients who had male gender, FSGS, post-transplant status, were children aged below 18 years, and had previously failed treatment. These factors were associated with a lower remission rate, although the correlations did not attain statistical significance. This underscores the diverse mechanisms that contribute to the development of primary FSGS.

87-89. Our hypothesis is that patients with primary FSGS may possess underlying changes in adhesion molecules or extracellular matrix glycoproteins that are associated with previously unreported mutations that can be investigated using NGS.

188-191. In almost all cases (74%; n=17), classical or NOS (not otherwise specified) FSGS was the predominant morphological variant observed on biopsy. Perihilar variant was identified in four patients (17%; n=4), while the cellular and tip variants were observed in one patient each.

208-209. The frequencies of variations in the different genes that we have analyzed are shown in Figure 1.

306-308. In addition to other signals, adhesion molecules and extracellular matrix glycoproteins are components of a network that involves redundant interactions between molecules.

  1. Lines 98-99: Short general characteristic of the whole cohort should be presented.

According to the recommendation, we added a brief overview of the entire cohort.

Main patient characteristics are shown in Table 1. Briefly, mean age was 53.96 ± 18.81 years, 26% were female and 74% suffered high blood pressures. Mean estimated glomeru-lar filtration rate was 60.30 ± 29.95 ml/min, mean proteinuria was 10.13 ± 5.96 g/day, and 43% showed nephrotic syndrome.

  1. Line 125, 126, etc: Reference and a description “In brief” should be presented for each of the methods used in the study.
  2. Lines 129,132,138, 147, 156, 159: For each kit or reagent the authors should indicate the Company, City, and Country where it was purchased.

(4 and 5) We rewrite the methods, adding recommendations in each case.

  1. Lines 188-190: The detailed analysis of patients’ clinical data should be presented. The authors should also describe the differences between different SKD stages.

Patients’ clinical and analytical data are shown in table 1. 

We add the staging classification of Chronic Kidney Disease according to the established KDIGO classification.

  1. Lines 218-221: The authors should describe specific details of 2 patients with ITGB and 5 patients with CDH1 where the most important mutations were found.

We develop the characteristics and evolution of patients with the referred mutations.

  1. Line 266: It is not clear which COL4A gene and collagen are described. This should be corrected.

COL4A (A3/A4/A5)

  1. Limitations cannot be considered as Conclusions. Limitations should be moved to the Discussion section.

Limitations are moved to the Discussion section.

  1. Conclusions are missing. Conclusions should describe the authors findings.

The conclusion is written according to the recommendation, with the findings of the study.

Round 3

Reviewer 2 Report

Comments

1.      Lines 141-142: This sentence is not clear. It should be clarified.

2.      Previous comment #5 has not been completely addressed: For each kit or reagent the authors should indicate the Company, City, and Country where it was purchased. This should be corrected.

3.      Reference 1 should be completed.

Lines 141-142: This sentence is not clear. It should be clarified.

Author Response

  1. Lines 141-142: This sentence is not clear. It should be clarified.

The sentence is rewritten for better understanding.

  1. Previous comment #5 has not been completely addressed: For each kit or reagent the authors should indicate the Company, City, and Country where it was purchased. This should be corrected.

As far as we understand, according to citation rules, for each kit or reagent, the Company, City, and Country where it was acquired should be indicated the first time it appears in the text, and subsequently, if it appears again, only the name should be used. This is the case with ThermoFisher in this instance. However, we can add it if you consider it necessary, but, if this is not what you are referring to, could you clarify it for us?

  1. Reference 1 should be completed.

Reference 1 is completed.
